# Green Walkability and Physical Activity in UK Biobank: A Cross-Sectional Analysis of Adults in Greater London

**DOI:** 10.3390/ijerph19074247

**Published:** 2022-04-02

**Authors:** Charlotte Roscoe, Charlotte Sheridan, Mariya Geneshka, Susan Hodgson, Paolo Vineis, John Gulliver, Daniela Fecht

**Affiliations:** 1MRC Centre for Environment and Health, Imperial College London, St Mary’s Hospital, Praed Street, London W2 1NY, UK; charlotte.sheridan2@gmail.com (C.S.); dr.susan.hodgson@gmail.com (S.H.); p.vineis@imperial.ac.uk (P.V.); d.fecht@imperial.ac.uk (D.F.); 2Department of Environment and Health, Harvard T.H. Chan School of Public Health, Harvard University, Landmark Center, Floor 3 West, 401 Park Drive, Boston, MA 02215, USA; 3London School of Hygiene & Tropical Medicine, Keppel St., London WC1E 7HT, UK; 4Department of Health Sciences, University of York, York YO10 4DD, UK; mmg529@york.ac.uk; 5Centre for Environmental Health and Sustainability, School of Geography, Geology and the Environment, University of Leicester, Leicester LE1 7LW, UK; jg435@le.ac.uk

**Keywords:** greenspace, walkability, street trees, physical activity, accelerometer, moderate-vigorous physical activity, walking, network buffer, urban planning, prospective cohort

## Abstract

Urban greenspace provides opportunities for outdoor exercise and may increase physical activity, with accompanying health benefits. Areas suitable for walking (walkability) are also associated with increased physical activity, but interactions with greenspace are poorly understood. We investigated associations of *walkability* and *green walkability* with physical activity in an urban adult cohort. We used cross-sectional data from Greater London UK Biobank participants (*n* = 57,726) and assessed *walkability* along roads and footpaths within 1000 m of their residential addresses. Additionally, we assessed *green walkability* by integrating trees and low-lying vegetation into the walkability index. Physical activity outcomes included self-reported and accelerometer-measured physical activity and active transport. We assessed associations using log-linear, logistic and linear regression models, adjusted for individual- and area-level confounders. Higher *green walkability* was associated with favourable International Physical Activity Questionnaire responses and achievement of weekly UK government physical activity guideline recommendations. Participants living in the highest versus lowest quintile of *green walkability* participated in 2.41 min (95% confidence intervals: 0.22, 4.60) additional minutes of moderate-and-vigorous physical activity per day. Higher *walkability* and *green walkability* scores were also associated with choosing active transport modes such as walking and cycling. Our *green walkability* approach demonstrates the utility in accounting for walkability and greenspace simultaneously to understand the role of the built environment on physical activity.

## 1. Introduction

Regular physical activity reduces the risk of non-communicable diseases, including cardiovascular disease, diabetes, and breast and bowel cancer, and protects against important disease risk factors such as overweight and obesity [1,2]. In 2018, the WHO published a global action plan on physical activity with the aim of increasing physical activity levels for all people [3]. The plan contained strategic objectives for planning active environments, which included: (a) strengthening access to good-quality, public, green and open spaces and green networks; and (b) enabling walking, cycling and other forms of active mobility via well-connected, compact neighbourhood design, with access to services [3]. These objectives suggest that physical activity levels can be improved by increasing access to greenspace and improving neighbourhood walkability.

Greenspace, land predominantly covered by vegetation, is hypothesised to improve levels of physical activity by providing accessible outdoor space that is suitable for exercise, with accompanying health benefits [4,5]. Reviews of the greenspace and health literature suggest a beneficial association of greenspace and physical activity yet highlight inconsistencies [4,5,6]. A seminal study on greenspace and physical activity in the Netherlands, for example, showed no association of greenspace with achieving national physical activity recommendations [7]. Klompmaker and colleagues, however, showed that vegetation cover within small (300 m) circular distance buffers of residential addresses was associated with increased odds of self-reported outdoor physical activity in the highest compared to the lowest quintile of vegetation cover (odds ratio (OR) 1.14; 95% confidence intervals (CIs) 1.10, 1.17) [8]. Additionally, a study at the English lower super output area (LSOA; average population ~1500) level, showed associations of greenspace cover with physical activity; however, positive associations were for non-recreational physical activity (e.g., do it yourself) and the study showed negative associations of greenspace cover with walking physical activity [9].

Neighbourhood walkability is usually estimated using a walkability index, which incorporates population density, the density of potential destinations (including shops, transport stops, and public amenities), and the density of the street networks that connect them [10]. As such, green spaces suitable for recreational physical activity (e.g., public parks) are characteristically low-density environments, which score low on walkability indices. Number of parks was included in the International Physical Activity and the Environment Network (IPEN) study walkability index, which was associated with walking for transport and for leisure [11], and with objectively measured moderate-and-vigorous physical activity (MVPA) [12]. Stockton and colleagues developed a walkability index specifically for London, UK, using physical activity data from an occupational cohort of older adults, and showed that higher density-based walkability in the areas where participants resided was associated with achieving more than 6 h of walking per week (OR 1.40; 95% CIs 1.1, 1.9); however, the addition of public recreational spaces, which included public parks, to the index did not substantially alter associations with walking [13].

For city planners, balancing neighbourhood density with adequate access to greenspace is a challenge, as open green spaces (e.g., parks) contain few residences, destinations, and street networks, which contribute to walkability. One exception is street trees, which can be integrated into a dense, compact urban plan that supports walking. Respondents to the London Travel Demand Survey showed higher odds of walking associated with a higher density of street trees around their residential address (OR = 1.06, 95% CIs = 1.03, 1.10); street connectivity was also positively associated with walking, but total vegetation cover surrounding addresses was not associated with walking [14].

Few environmental epidemiological studies have reported on the spatial relationship of neighbourhood walkability and greenspace [15], though some studies showed negative correlations [9,16,17,18,19,20]. For example, a study on multiple urban environmental exposures found a non-linear, negative association of vegetation cover and walkability surrounding residential addresses of US-based Nurses Health Study cohort participants [16]. Additionally, using the UK Biobank cohort study, a positive association of greenspace and self-reported walking of more than 30 min per day was found, which remained after adjustment for density-based walkability metrics [21]. Mutual adjustment, however, was not explored in this analysis, and the individual impact of walkability was not reported [21]. In 2022, protective associations of higher residential greenness and higher walkability with lower arterial stiffness were found using mutually adjusted models, though mediation by physical activity was not assessed [20].

Walkability and greenspace jointly exert influences on physical activity in real-world settings. They are not only spatially correlated, but are experienced as a single, integrated exposure that may encourage or discourage physical activity during the commute (e.g., active travel) and during leisure time. Further exploration of the relationship of greenspace and walkability, and their combined impact on physical activity, is crucial to guide strategies for active neighbourhood planning. More specifically, the different impact of trees, which can be planted along streets without substantially impacting business and residential density (i.e., components of walkability), compared to other types of vegetation (e.g., grass), is essential information for greening cities to encourage physical activity. We aimed to integrate vegetation cover into a *walkability* index to create a *green walkability* index to explore the added benefit of green space exposure on self-reported and objectively measured physical activity, using UK Biobank. Our hypothesis was that *green walkability* has added benefits compared to *walkability* in terms of increasing physical activity.

## 2. Materials and Methods

### 2.1. Study Population

We used cross-sectional data from UK Biobank, a large cohort of volunteer participants aged between 40 and 69 years at recruitment in the period 2006–2010 [22]. Participants completed baseline lifestyle questionnaires, assisted in physical tests, and provided residential addresses. At the time of our analysis, data were available on 502,656 participants across Great Britain. We used information from 57,726 participants that were enrolled at UK Biobank Assessment Centres within the Greater London administrative boundary (Barts, Croydon or Hounslow) for which we had spatially high-resolution information on height-attributed vegetation cover available. Additionally, a subset participated in a 7 day accelerometer-based physical activity follow-up (*n* = 14,095). After we excluded participants defined by UK Biobank as having poor accelerometer wear time (*n* = 1108) and failed accelerometer calibration (*n* = 5), 12,986 participants who had exposure data and accelerometer data remained for our analysis. UK Biobank has ethics approval from the North West Multi-centre Research Ethics Committee (reference 16/NW/0274).

### 2.2. Self-Reported Physical Activity

Weekly average physical activity levels and achievement of the UK government’s physical activity recommendations were available from UK Biobank. UK Biobank derived these variables from participants’ responses to the UK Biobank baseline questionnaire, which included the short-form International Physical Activity Questionnaire (IPAQ). Weekly physical activity levels were categorised as low, moderate, and high following the IPAQ protocol [23]. In brief, UK Biobank calculated categories of IPAQ physical activity using total volume and number of days/sessions of physical activity to capture regular participation in physical activity across all domains (leisure, domestic, work-related and transport physical activity).

Metabolic equivalent of task (MET)-minutes were based on each participant’s IPAQ responses taking into account the metabolic intensity of an activity compared to sitting quietly. Following IPAQ convention, values were represented as a ratio for sitting versus walking (3.3 METs), sitting versus moderate physical activity (4.0 METs) and sitting versus vigorous physical activity (8.0 METs). UK Biobank multiplied MET-minutes by the number of days an activity was conducted per week. High physical activity corresponded to: (a) vigorous-intensity physical activity on at least 3 days, achieving a minimum total physical activity of at least 1500 MET-minutes per week; or (b) 7 or more days of any combination of walking, moderate- or vigorous-intensity physical activity, achieving a minimum total physical activity of at least 3000 MET-minutes per week. Moderate physical activity corresponded to: (a) 3 or more days of vigorous-intensity physical activity of at least 20 min per day; or (b) 5 or more days of moderate-intensity physical activity and/or walking of at least 30 min per day; or (c) 5 or more days of any combination of walking, moderate-intensity or vigorous-intensity physical activity, achieving a minimum total physical activity of at least 600 MET-minutes per week. Low physical activity corresponded to any level of physical activity below the moderate category threshold.

Based on responses to the IPAQ short form, UK Biobank also provided a binary outcome variable for achieved versus not-achieved UK physical activity recommendations (more than 150 MET-minutes per week) calculated using (a) moderate-and-vigorous physical activity (MVPA); and (b) MVPA and walking [24].

### 2.3. Accelerometer Measured 7 Day Physical Activity

Between February 2013 and December 2015, participants who had provided a valid email address were randomly selected and invited to participate in an accelerometer follow-up study. UK Biobank conducted the accelerometer-based follow-up in the volunteer sample between 2013 and 2016 to objectively measure physical activity over a 7 day period; detailed methodology is described elsewhere [25]. In brief, participants were mailed wrist-worn accelerometers and asked to wear them immediately and continuously on their dominant (writing) hand for one week. Physical activity was extracted by UK Biobank from 100 Hz raw triaxial acceleration data after calibration, removal of gravity and sensor noise, and identification of non-wear episodes [25]. UK Biobank provided 7 day accelerometer physical activity estimates as fraction of time during which acceleration intensity was at or below various threshold values. We used these intensity summaries to create moderate, vigorous, and MVPA minutes per day using thresholds of 125–425 milli-gravities (mg), more than 425 mg, and more than 125 mg, respectively, which corresponds to prior UK Biobank accelerometer research [26,27]. We derived intensity estimates in minutes per day above specified thresholds (e.g., MVPA minutes per day as 24 × 60× (1-fraction acceleration ≤125 mg)) and minutes per day between specified thresholds (e.g., moderate physical activity minutes per day as 24 × 60 × (fraction acceleration >425 mg)-(24 × 60 × (fraction acceleration >125 mg)). As estimation of physical activity from accelerometer data is sensitive to intensity thresholds [28], we assessed the impact of using different activity thresholds (±25 mg) in sensitivity analyses.

### 2.4. Transport Mode Choice for Commute and Non-Commute Travel

We used the baseline questionnaire responses on type of transport used to and from work, and on trips other than for work, as indicators of commute and non-commute travel choice, categorised into car/motor vehicle, walking, public transport, and cycling. We also obtained information on distance between home and work as a potential confounder for active commuting.

### 2.5. Exposure Assessment

#### 2.5.1. Walkable Network Buffers

We used participants’ geocoded residential addresses at baseline to assess walkability in a 1000 m network buffer of study participants’ addresses. The use of a 1000 m network buffer was informed by prior examples in the literature [11,29,30]. Network buffers capture features within walking distance more accurately than circular distance buffers [30]. To capture walkable roads and pedestrianised routes, we integrated Ordnance Survey (OS) Integrated Transport Network (ITN), excluding motorways, with the OS Urban Paths Theme (UPT) network extension. OS datasets were provided by Edina via Digimap [31].

We used pgRouting and PostGIS (v. 2.3.3) routing and spatial extenders for the open-source database software Postgres (v. 9.6) to construct the network buffers. We modified a version of the pgRouting function *pgr_withPointsDD* to trace the road/path network lines 1000 m in all directions starting from the point on the routed network closest to each participant’s address. We added a 50 m width buffer around each residential 1000 m line-based network to create a walkable network polygon buffer (Figure 1), within which we assessed *walkability* and *green walkability*.

#### 2.5.2. Walkability

We used a density-based approach to derive *walkability*, similar to Stockton and colleagues [13]. We combined domains of walkability relating to (i) population density, (ii) street connectivity and (iii) destination density, to model *walkability*. To estimate population density, we used Office for National Statistics (ONS) estimates of number of residents per residential postcode centroid (representing on average of 12 households), based on 2011 census data [32], corresponding to the UK Biobank population at baseline (2006–2010). We spatially joined postcode centroids to walkable network buffers, and calculated population density by dividing the population by buffer area. To estimate street connectivity, we used OS ITN with OS UPT to represent roads with vehicle access, as well as footpaths, cycle paths and other pedestrianised throughways. We calculated the number of 3-way (or more) intersections (i.e., junctions) along the routed network within each 1000 m network buffer as a measure of route connectivity. To estimate destination density, we used OS points of interest (POI), which contain approximately 4 million unique geographical features across Great Britain, including businesses, services, transport, and public infrastructure. POIs have coordinates with a precision of less than 1 m and functional categories. We used functional categories to select and retain POIs that we deemed relevant for walkability assessment. Relevant POIs included restaurants, shops, markets, banks, sports facilities, hair and beauty services, schools, health centres, post offices, libraries, places of worship, public transport stations and bus stops (see Appendix A for full list of categories). Each of the three domains of walkability were converted to *z*-scores and summed with equal weighting to create a walkability *z*-score.

#### 2.5.3. Green Walkability

We used vegetation information from The GeoInformation Group [33] to develop a measure of *green walkability* for Greater London at 2.5 m × 2.5 m resolution, by integrating vegetation information into the *walkability* index. Data were derived from a combination of aerial imagery, Light Detection and Ranging (LiDAR) data and manually digitised tree data. Vegetation data were categorised by height into tree cover (equal to or higher than 2.5 m) and ground cover (lower than 2.5 m) (Figure 1). In this study, ground cover is a catch-all term that we use to describe ground vegetation, shrubs, and saplings. The granularity of the available greenspace data was our principal motivation for focusing on UK Biobank participants residing in Greater London.

To integrate greenspace into the walkability index, we joined the vegetation data to the walkable 1000 m network buffers for each participant’s address, and calculated percentage vegetation cover of the total network buffer area. To ensure that the vegetation data fully covered the network distance buffers around each participant’s address geocode, we included only addresses located 1000 m inwards of the vegetation data extent for all walkability scores. We converted percentage cover into a *z*-score and added this to the 3 walkability domain *z*-scores described above. We weighted the 4 domains of the *green walkability* score equally. We repeated this process for tree cover and ground cover, separately, to produce the *tree-cover walkability* and *ground-cover walkability*.

### 2.6. Statistical Analysis

We used multinomial log-linear models to assess associations of UK Biobank participant weekly average physical activity levels, as reported on the short form IPAQ at baseline, using Low physical activity as the reference category (*n* = 57,726). We fitted logistic regression models to assess if the likelihood of achieving UK recommended physical activity levels via MVPA was associated with *walkability* and *green walkability* (*n* = 57,726). We repeated this analysis using MVPA and low-intensity physical activity contributions (i.e., walking) combined towards achieving UK physical activity recommendations. We used linear regression models to assess associations of accelerometer measured minutes of moderate, vigorous, and MVPA with quintiles of *walkability* and *green walkability* (*n* = 12,986). We assessed the impact of using different activity thresholds (± 25 mg) in sensitivity analyses. We used multinomial log-linear models to assess associations of *walkability* and *green walkability* with transport mode used for commuting to the workplace using car/motor vehicle transport as the reference category (*n* = 23,999). This analysis was repeated for non-commute transportation (*n* = 44,998). All models were adjusted for participant’s age at baseline, sex, average household income before tax, and the area-based Index of Multiple Deprivation (IMD 2015) [34], a composite measure of neighbourhood deprivation. Models for commuting transport mode were additionally adjusted for distance from home address to workplace. We exponentiated the coefficients from models to obtain odds (risk ratio) compared to reference category odds of 1 and calculated 95% confidence intervals. To improve interpretability and account for non-linear associations, we used quintiles of *walkability* and *green walkability*, with quintile 1 (least walkable 20% of participants’ addresses) as the comparative reference category across analyses. Analyses were conducted in R (v. 4.0.1) via R Studio (v. 1.3.959) using *nnet* (https://cran.r-project.org/web/packages/nnet/nnet.pdf; accessed on 27 February 2022) and base R packages.

## 3. Results

Study participants (*n* = 57,726) were on average 56 years old at baseline (2006–2010), 55.7% female, and lived predominately in low or medium deprivation areas (75.2%). Participants with additional accelerometer information (*n* = 12,986) were more likely to live in low and medium deprivation areas (81.1%) and in high-income households with an annual income of more than £52,000 before tax (40.6%). Participants with accelerometer information also reported slightly higher levels of physical activity at baseline. Participants with commute travel information who commuted at least once per week (*n* = 23,999) were younger, more likely to earn over £100,000 and less likely to earn under £18,000 average total household income before tax (Table 1). Participant characteristics by quintiles of exposure can be found in the Appendix A.

Overall, we observed a positive association of higher physical activity with increasing *walkability* and *green walkability* compared to less walkable areas (Table 2). UK Biobank participants residing in the most walkable areas (quintile 5) had a 51% (95% CIs: 36%, 67%) and 45% (95% CIs: 31%, 61%) increased odds of moderate and high physical activity, respectively, compared to those in the lowest walkable areas (quintile 1). Additionally, when dichotomised into those who achieved and did not achieve UK physical activity recommendation guidelines, participants living in more walkable areas had higher odds of achieving recommendations via MPVA and walking. Achieving UK physical activity recommendations was also associated with *walkability* and *green walkability* when only MVPA was used, following a similar trend to analyses that used MPVA and walking. *Green walkability* compared to *walkability* showed a similar pattern across all IPAQ self-reported measures, though associations were attenuated.

Using objectively measured (accelerometer-based) physical activity, we found that minutes per day spent on MVPA were higher for participants residing in the highest compared to the lowest quintile of *green walkability* (Table 3). This trend was largely driven by moderate physical activity differences. *Tree-cover walkability*, as opposed to either *green walkability* or *ground-cover walkability*, showed the largest increase in minutes per day of accelerometer-based MVPA in the most versus least walkable quintiles (3.64 min per day; 95% CIs: 1.45, 5.84).

Higher walkability was associated with higher odds of all forms of active transport and public transport compared to driving a motor vehicle for both commuting and non-commute purposes (Table 4). For example, participants residing in the most versus least walkable quintile were 5 times more likely (OR 5.06, 95% CI 4.46, 5.74) to walk for commuting and 6 times more likely (OR 6.37, 95% CI 5.85, 6.94) to walk for non-commuting purposes rather using a motor vehicle. Again, odds were slightly higher for the *walkability* compared to *green walkability* index.

## 4. Discussion

### 4.1. Summary of Findings

In our UK Biobank cross-sectional analysis, we found that both *walkability* and *green walkability* were associated with all measures of physical activity. This includes self-reported IPAQ physical activity levels and mode of transport as well as objectively measured physical activity indicators. We found that participants living in the most compared to the least walkable quintile, using either *walkability* or *green walkability*, had higher odds of achieving UK physical activity recommendations, after adjustment for individual and area-level covariates. Participants in the most compared to the least walkable quintiles were more likely to achieve these recommendations regardless of whether we assessed MVPA and walking, or only MVPA. Additionally, participants were more likely to use active modes of transport if they resided in the most walkable quintile versus the least walkable quintile. Using accelerometer-based measurements, we showed that UK Biobank participants residing in the most compared to the least walkable quintiles for *walkability* and *green walkability* undertook more minutes per day of moderate physical activity.

For self-reported measures, differences in effect estimates between the least-walkable and most-walkable quintiles were smaller for *green walkability* compared to *walkability*. That is, the addition of vegetation cover into the *walkability* index to produce *green walkability* index diluted associations with self-reported physical activity and self-reported active transport use. However, more minutes per day of objective (i.e., accelerometer-measured) MVPA were recorded in participants residing in the highest quintile of *green walkability* compared to the lowest quintile, and the difference between the most-walkable and least-walkable quintiles was larger for *green walkability* compared to *walkability*.

### 4.2. Integrating Walkability and Greenspace

To our knowledge, the integration of high-resolution data on vegetation into a walkability index has not previously been attempted. Other studies have integrated land use, including greenspace, into walkability indices [11], though they did not focus on the impacts of including and excluding vegetation cover in walkability indices on physical activity levels. Our study combined walkability and greenspace exposure assessment methods. We used, for example, a distance-based street network buffer to capture accessible destinations and integrated high-resolution vegetation cover within the network buffer. This approach allowed for an integrated assessment of co-existing exposures, with potentially synergistic effects in relation to physical activity, to be assessed via the *green walkability* index.

Our integrated exposure assessment approach is novel in an environmental epidemiological context. Major international guidelines, including the WHO global action plan on physical activity [3], cite studies that separately assessed walkability and greenspace exposures. Joint analyses of spatially correlated urban exposures, however, are necessary to understand the relative importance of urban exposures for physical activity to better guide urban development and planning. Given our findings, we echo the view of Rugel and Brauer [15] that walkability and greenspace should be considered jointly in epidemiological analyses. Moreover, we suggest that spatially correlated exposures can be explored in an integrated approach using exposure assessment techniques, as an alternative to treating spatially correlated exposures as confounding factors during analysis.

### 4.3. Findings in Context

Our *walkability index* was based on that of Stockton and colleagues [13], which was developed using an older-age adult cohort residing in Greater London (Whitehall II). We showed that a similar density-based walkability index was associated with self-reported physical activity levels in older adults who resided in Greater London at UK Biobank cohort baseline. Furthermore, we showed that participants living in the most compared to the least walkable quintile had higher odds of achieving UK physical activity recommendations, even when walking activity was excluded and only MVPA was considered. A study of walkability in multiple cities across five continents also showed that walkability was associated with objectively measured MVPA participation in a multi-country study, which included the UK [12]. Using accelerometer-based measurements, we found that UK Biobank participants residing in the most compared to the least walkable quintile of *green walkability* participated in more minutes of MVPA; furthermore, we showed that this was driven by moderate physical activity rather than vigorous physical activity in older adults.

A UK Biobank study showed that walking and cycling during the commute were associated with a lower risk of cardiovascular disease [35]. Initiatives to encourage and support active commuting could reduce the risk of chronic conditions. We found a strong dose–response association of *walkability* and active travel in UK Biobank participants residing in Greater London, additionally, we provide some evidence that *green walkability* is associated with overall objectively measured MVPA participation. The UK Chief Medical Officers’ guidelines recommend that older adults participate in a minimum of 150 min of moderate-intensity physical activity or 75 min of vigorous-intensity physical activity per week [24]. Compared to previous guidelines [36], the newer guidelines put a stronger emphasis on regular physical activity for older adults and removed the recommendation of a 10 min minimum session duration that was previously advised [24]. Evidence on the benefits of cumulative, low-to-moderate-intensity physical activity on likelihood of developing cardiovascular disease risk factors (e.g., diabetes) and premature mortality in older adults prompted these changes [37,38,39,40,41]. Based on our findings, we suggest that walkability is a crucial consideration for promoting regular, low-to-moderate-intensity physical activity, and vegetation cover surrounding walkable routes might increase participation in moderate physical activity.

### 4.4. Implication for Urban Planning

*Tree-cover walkability* was associated with the largest increase in accelerometer-measured MVPA in the most walkable compared to the least walkable areas. This is similar to findings from a study in London, UK, which showed that street network connectivity and street tree density surrounding residential addresses of respondents to the London Travel Demand Survey were associated with higher odds of walking [14]. For city planners balancing urban density with access to green spaces is a challenge, and an increase in street trees could provide a potential solution as they provide vegetation cover without compromising density. However, we suggest a cautious interpretation of *tree-cover walkability* results. The association of *tree-cover walkability* and moderate activity may be due to street trees providing a more appealing environment for moderate physical activity such as walking and, therefore, may be causally associated with more minutes of MVPA when integrated into a walkability index. However, as well as capturing tree cover, we acknowledge that street trees along major roads may, to some extent, double-count street connectivity in the *tree-cover walkability* index. Alternatively, the planting of trees along streets in historically wealthy neighbourhoods might offer a non-causal explanation to the association with physical activity, if participants residing in historically wealthy areas were more likely to exercise. To account for this, our analyses adjusted for small-area-level deprivation and household income, though we cannot exclude the potential of residual confounding by socio-economic factors on associations of all *walkability* exposures and physical activity.

### 4.5. Limitations

There are some limitations to our analysis. Firstly, analyses were cross-sectional, as we did not have longitudinal exposure and outcome data, which limits causal interpretation. There is also potential that more physically active individuals may have moved to more walkable and greener neighbourhoods and that increasing *walkability* or *green walkability* would not increase physical activity levels in individuals. Our analysis raises multiple future avenues of inquiry regarding *green walkability* and *tree-cover walkability* and their relationship with moderate physical activity, which requires assessment using repeated exposure and physical activity measurements. A study conducted using UK Biobank Stockport (Greater Manchester assessment centre), where repeat assessment data is available, showed modal shift in commute transport to walking and cycling was associated with lower body mass index and protective cardiovascular effects [35,42]. Further work on the UK Biobank Stockport subcohort or other cohorts with longitudinal accelerometer data would offer the opportunity to assess potential environmental drivers of modal shift (e.g., greening influences over time). Furthermore, we used equal weights to sum the domains of the walkability score. Applying different weights for vegetation cover compared to population, destination and junction density could alter associations of *green walkability* and physical activity, as suggested in other research [43]. Finally, the lower confidence intervals of accelerometer-based minutes of MVPA were close to zero (e.g., participants living in the highest versus lowest quintile of *green walkability* participated in 2.41 min (95% confidence intervals: 0.22, 4.60) additional minutes per day); however, even small increases in MVPA may have health benefits and environmental associations should be further explored in other cohorts. Future analyses should explore physical activity beyond commute versus non-commute activity and may benefit from understanding differences in leisure-time, occupational and domestic physical activity in relation to greenspace and walkability.

## 5. Conclusions

Balancing walkability and greenspace exposure, which are inversely related in space, is an important urban planning challenge with large health implications. Researchers can integrate exposure assessments to explore potential solutions that improve physical activity levels. Our *green walkability* approach demonstrates the utility of accounting for greenspace and walkability, jointly, in physical activity research. Further, the comparison between types of *green walkability*, including *ground-cover* and *tree-cover walkability*, and standard *walkability* serves to explore specific greenspace measures within the context of walkable networks.

## Figures and Tables

**Figure 1 ijerph-19-04247-f001:**
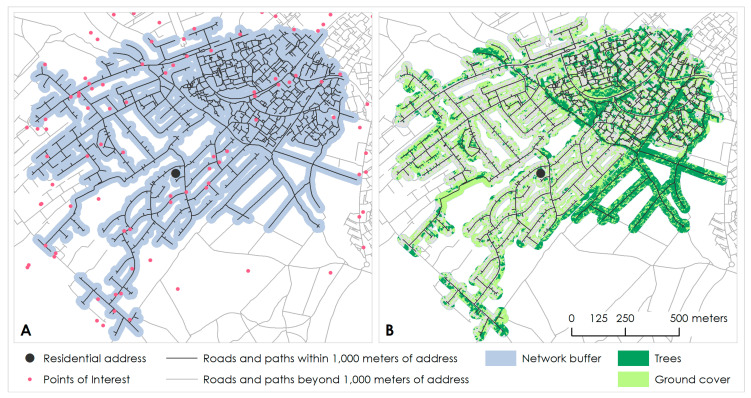
Example of a road/path network within 1000 m of a (dummy) cohort participant residential address, with a 50 m width buffer of the road/path network. Panel (**A**) shows points of interest (e.g., shops and public transport stops; pink) and panel (**B**) shows trees (dark green) and ground cover (light green) in the network buffer.

**Table 1 ijerph-19-04247-t001:** Baseline characteristics of UK Biobank participants (*n* = 57,726), participants with 7 day accelerometer information (*n* = 12,986), participants with non-commute travel information (*n* = 44,998), and participants with commute travel information (*n* = 23,999); all study participants resided within Greater London administrative boundaries at study baseline (2006–2010).

	Main Study Participants (*n* = 57,726)	Participants with Accelerometer Information (*n* = 12,986)	Participants with Non-Commute Travel Information (*n* = 44,998)	Participants with Commute Travel Information (*n* = 23,999)
Age at baseline (years)				
Mean (SD)	56.0 (8.23)	56.1 (8.01)	56.0 (8.26)	52.7 (7.34)
Sex				
Female	32,141 (55.7%)	7672 (59.1%)	25,015 (55.6%)	12,925 (53.9%)
Male	25,585 (44.3%)	5314 (40.9%)	19,983 (44.4%)	11,074 (46.1%)
Average total household income before tax (£)				
Less than 18,000	8925 (15.5%)	1309 (10.1%)	6907 (15.3%)	1830 (7.6%)
18,000 to 30,999	10,106 (17.5%)	2162 (16.6%)	7891 (17.5%)	3985 (16.6%)
31,000 to 51,999	11,933 (20.7%)	3054 (23.5%)	9392 (20.9%)	5962 (24.8%)
52,000 to 100,000	12,044 (20.9%)	3480 (26.8%)	9351 (20.8%)	6643 (27.7%)
Greater than 100,000	5794 (10.0%)	1748 (13.5%)	4500 (10.0%)	3219 (13.4%)
Missing	8924 (15.5%)	1233 (9.5%)	6957 (15.5%)	2360 (9.8%)
Index of Multiple Deprivation (small-area level)				
Low deprivation	18,239 (31.6%)	4823 (37.1%)	14,256 (31.7%)	7191 (30.0%)
Medium deprivation	25,191 (43.6%)	5740 (44.2%)	19,582 (43.5%)	10,667 (44.4%)
High deprivation	14,296 (24.8%)	2423 (18.7%)	11,160 (24.8%)	6141 (25.6%)
IPAQ activity group				
Low	8055 (14.0%)	1713 (13.2%)	6229 (13.8%)	3419 (14.2%)
Moderate	21,011 (36.4%)	5041 (38.8%)	16,508 (36.7%)	9059 (37.7%)
High	18,934 (32.8%)	4594 (35.4%)	14,744 (32.8%)	7981 (33.3%)
Missing	9726 (16.8%)	1638 (12.6%)	7517 (16.7%)	3540 (14.8%)
Achieved UK physical activity recommendations via MVPA and walking ^1^				
Yes	40,050 (69.4%)	9652 (74.3%)	31,330 (69.6%)	16,984 (70.8%)
No	7936 (13.7%)	1696 (13.1%)	6138 (13.6%)	3471 (14.5%)
Missing	9740 (16.9%)	1638 (12.6%)	7530 (16.7%)	3544 (14.8%)
Achieved UK physical activity recommendations via MVPA only ^1^				
Yes	25,975 (45.0%)	6294 (48.5%)	20,222 (44.9%)	10,657 (44.4%)
No	22,025 (38.2%)	5054 (38.9%)	17,259 (38.4%)	9802 (40.8%)
Missing	9726 (16.8%)	1638 (12.6%)	7517 (16.7%)	3540 (14.8%)
7 day accelerometer: minutes spent doing MVPA (≥125 mg)				
Mean (SD)	-	75.8 (36.1)	-	-
7 day accelerometer: minutes spent doing moderate activity (≥125 mg and <425 mg)				
Mean (SD)	-	71.9 (33.4)	-	-
7 day accelerometer: minutes spent doing vigorous activity (≥425 mg)				
Mean (SD)	-	3.93 (5.62)	-	-
Non-commuting transport				
Car/motor vehicle	25,084 (43.5%)	5764 (44.4%)	25,084 (55.7%)	14,021 (58.4%)
Cycle	1017 (1.8%)	296 (2.3%)	1017 (2.3%)	676 (2.8%)
Walk	11,931 (20.7%)	2929 (22.6%)	11,931 (26.5%)	5761 (24.0%)
Public transport	6966 (12.1%)	1160 (8.9%)	6966 (15.5%)	3474 (14.5%)
Missing	12,728 (22.1%)	2815 (21.7%)	0 (0%)	67 (0.3%)
Commuting transport ^2^				
Car/motor vehicle	11,722 (20.3%)	2576 (19.8%)	11,679 (26.0%)	11,374 (47.4%)
Cycle	1094 (1.9%)	326 (2.5%)	1093 (2.4%)	1077 (4.5%)
Walk	4748 (8.2%)	1340 (10.3%)	4740 (10.5%)	4639 (19.3%)
Public transport	7119 (12.3%)	1570 (12.1%)	7098 (15.8%)	6909 (28.8%)
Missing	33,043 (57.3%)	7174 (55.3%)	20,388 (45.3%)	0 (0%)

^1^ UK physical activity guideline recommendations (150 MET minutes per week) [24]. MVPA = moderate-vigorous physical activity. ^2^ Participants who commute to work at least once per week.

**Table 2 ijerph-19-04247-t002:** Odds (risk ratio) and 95% confidence intervals (CIs) of high and moderate compared low physical activity levels based on self-reported International Physical Activity Questionnaire (IPAQ) responses by quintiles of walkability (Q1 is the lowest 20% of walkability across participants’ addresses and is used as reference category; Q5 represents the highest 20% of walkability) (*n* = 57,726). Walkability, green walkability, ground-cover walkability and tree-cover walkability were assessed as the independent (exposure) variable in separate models (*n* = 57,726 for all models). Odds and 95% confidence intervals of achieving UK physical activity guidelines (150 MET minutes per week) based on MVPA only, and MVPA and walking, by quintiles of walkability, green walkability, ground-cover walkability and tree-cover walkability are also shown. All models are adjusted for: age, sex, average household income before tax, and area-level deprivation. Bold font indicates *p* ≤ 0.05.

Self-Reported Weekly Physical Activity	Walkability				Green Walkability				Ground-Cover Walkability				Tree-Cover Walkability			
	Q2	Q3	Q4	Q5	Q2	Q3	Q4	Q5	Q2	Q3	Q4	Q5	Q2	Q3	Q4	Q5
IPAQ activity group: Moderate ^1^	**1.13** **(1.03, 1.23)**	**1.21** **(1.11, 1.33)**	**1.28** **(1.17, 1.41)**	**1.51** **(1.36, 1.67)**	**1.1** **(1.02, 1.22)**	**1.13** **(1.03, 1.23)**	**1.18** **(1.08, 1.29)**	**1.39** **(1.26, 1.53)**	1.07 (0.98, 1.17)	**1.15** **(1.05, 1.26)**	**1.11** **(1.01, 1.21)**	**1.34** **(1.21, 1.47)**	1.03 (0.94, 1.12)	**1.13** **(1.04, 1.24)**	**1.24** **(1.13, 1.35)**	**1.41** **(1.28, 1.56)**
IPAQ activity group: High ^1^	**1.15** **(1.05, 1.25)**	**1.21** **(1.10, 1.32)**	**1.23** **(1.12, 1.36)**	**1.45** **(1.31, 1.61)**	1.10 (1.00, 1.20)	**1.15** **(1.05, 1.25)**	**1.21** **(1.11, 1.33)**	**1.33** **(1.20, 1.46)**	1.06 (0.97, 1.16)	**1.11** **(1.01, 1.21)**	**1.12** **(1.02, 1.23)**	**1.26** **(1.14, 1.39)**	1.02 (0.93, 1.11)	**1.17** **(1.07, 1.28)**	**1.20** **(1.10, 1.31)**	**1.36** **(1.24, 1.51)**
Achieved national recommendations via MVPA and walking ^2^	**1.13** **(1.04, 1.23)**	**1.22** **(1.12, 1.32)**	**1.31** **(1.20, 1.43)**	**1.48** **(1.35, 1.63)**	1.08 (0.99, 1.17)	**1.11** **(1.03, 1.21)**	**1.22** **(1.12, 1.33)**	**1.36** **(1.24, 1.49)**	1.04 (0.96, 1.13)	**1.11** **(1.02, 1.20)**	**1.15** **(1.05, 1.25)**	**1.29** **(1.18, 1.41)**	1.05 0.97, 1.14)	**1.16** **(1.07, 1.25)**	**1.26** **(1.16, 1.37)**	**1.42** **(1.30, 1.56)**
Achieved national recommendations via MVPA ^2^	1.04 (0.98, 1.10)	1.04 (0.98, 1.11)	1.03 (0.97, 1.10)	**1.09** **(1.02, 1.17)**	1.02 (0.96, 1.08)	1.04 (0.98, 1.11)	1.05 (0.98, 1.12)	**1.09** **(1.02, 1.16)**	1.02 (0.96, 1.08)	1.02 (0.96, 1.08)	1.03 (0.96, 1.09)	1.07 (1.00, 1.14)	0.97 (0.92, 1.04)	1.05 (0.99, 1.12)	1.01 (0.95, 1.08)	1.07 (1.00, 1.14)

^1^ Reference category low physical activity group derived from the International Physical Activity Questionnaire (IPAQ). ^2^ Reference category did not achieve UK physical activity recommendations (>150 MET minutes per week) derived from IPAQ responses.

**Table 3 ijerph-19-04247-t003:** Effect estimates and 95% confidence intervals (CIs) for physical activity levels across quintiles of walkability scores, with quintile 1 (Q1) being the lowest 20% of a walkability score; quintile 5 (Q5) being the highest (*n* = 12,986). Effect estimates are presented as additional time in minutes spent participating in moderate physical activity, vigorous physical activity and in moderate-and-vigorous physical activity (MVPA), compared to the time (minutes per day) spent participating in those activities for participants residing in the reference category of walkability (Q1; lowest walkability). Models were adjusted for: age, sex, average household income before tax, and area-level deprivation. Bold font indicates *p* ≤ 0.05.

Accelerometer-Measured 7 Day Physical Activity	Walkability				Green Walkability				Ground-Cover Walkability				Tree-Cover Walkability			
	Q2	Q3	Q4	Q5	Q2	Q3	Q4	Q5	Q2	Q3	Q4	Q5	Q2	Q3	Q4	Q5
Moderate activity (125–425 mg)	0.14 (−1.73, 2.01)	0.34 (−1.56, 2.25)	1.43 (−0.52, 3.38)	**2.38** **(0.23, 4.54)**	0.48 (−1.39, 2.34)	0.93 (−0.95, 2.81)	**1.91** **(0.02, 3.80)**	**2.26** **(0.23, 4.30)**	0.03 (−1.84, 1.90)	−0.76 (−2.66, 1.14)	−0.04 (−1.90, 1.99)	0.79 (−1.25, 2.84)	1.21 (−0.65, 3.07)	1.85 (−0.01, 3.72)	**2.51** **(0.63, 4.39)**	**3.66** **(1.63, 5.70)**
Vigorous activity (>425 mg)	0.03 (−0.30, 0.35)	−0.07 (−0.40, 0.26)	0.12 (−0.22, 0.45)	−0.12 (−0.49, 0.26)	0.27 (−0.06, 0.59)	0.03 (−0.30, 0.36)	**0.410** **(0.08, 0.74)**	0.15 (−0.20, 0.50)	0.12 (−0.21, 0.44)	−0.06 (−0.39, 0.27)	0.17 (−0.17, 0.51)	0.00 (−0.35, 0.36)	0.03 (−0.29, 0.35)	0.00 (−0.33, 0.32)	0.23 (−0.10, 0.56)	0.00 (−0.38, 0.33)
MVPA (>125 mg)	0.16 (−1.85, 2.18)	0.27 (−1.78, 2.32)	1.55 (−0.55, 3.65)	2.27 (−0.05, 4.59)	0.7 4 (−1.26, 2.75)	0.96 (−1.06, 2.99)	**2.32** **(0.28, 4.36)**	**2.41** **(0.22, 4.60)**	0.15 (−1.87, 2.16)	−0.83 (−2.87, 1.22)	0.21 (−1.88, 2.31)	0.80 (−1.40, 3.00)	1.24 (−0.77, 3.25)	1.85 (−0.16, 3.86)	**2.74** **(0.71, 4.77)**	**3.64** **(1.45, 5.84)**

**Table 4 ijerph-19-04247-t004:** Odds (risk ratio) and 95% confidence intervals (CIs) of active commute transport mode and active non-commute transport mode compared to using car/motor vehicle for commute (*n* = 23,999) and non-commute (*n* = 44,998) transport shown by quintiles of *walkability* and *green walkability* (Q1 being the lowest 20% of walkability scores; Q5 the highest 20%). Models were adjusted for: age, sex, average household income before tax, area-level deprivation level and distance between home and workplace (commute only). Bold font indicates *p* ≤ 0.05.

Transport Mode (Ref = Car/Motor Vehicle)	Walkability				Green Walkability			
	Q2	Q3	Q4	Q5	Q2	Q3	Q4	Q5
Commute								
Cycle	**1.58 (1.25, 1.99)**	**1.92 (1.52, 2.42)**	**2.93 (2.33, 3.69)**	**6.69 (5.29, 8.46)**	**1.63 (1.28, 2.08)**	**2.14 (1.69, 2.71)**	**2.99 (2.37, 3.76)**	**5.77 (4.57, 7.27)**
Public transport	**1.35 (1.22, 1.48)**	**1.79 (1.63, 1.97)**	**2.74 (2.48, 3.02)**	**4.03 (3.61, 4.50)**	**1.25 (1.14, 1.38)**	**1.65 (1.50, 1.81)**	**2.09 (1.90, 2.29)**	**2.92 (2.64, 3.24)**
Walk	**1.28 (1.14, 1.44)**	**1.70 (1.52, 1.91)**	**2.52 (2.24, 2.83)**	**5.06 (4.46, 5.74)**	**1.25 (1.11, 1.40)**	**1.65 (1.47, 1.85)**	**2.20 (1.97, 2.47)**	**3.64 (3.23, 4.11)**
Non-commute								
Cycle	**1.43 (1.13, 1.80)**	**2.09 (1.68, 2.62)**	**2.66 (2.13, 3.33)**	**6.54 (5.21, 8.20)**	**2.13 (1.67, 2.72)**	**2.42 (1.90, 3.09)**	**3.30 (2.60, 4.18)**	**6.77 (5.34, 8.59)**
Public transport	**1.51 (1.37, 1.67)**	**2.00 (1.81, 2.20)**	**2.67 (2.42, 2.95)**	**4.09 (3.68, 4.55)**	**1.47 (1.33, 1.62)**	**1.99 (1.81, 2.19)**	**2.33 (2.12, 2.56)**	**3.28 (2.96, 3.63)**
Walk	**1.66 (1.53, 1.79)**	**2.47 (2.29, 2.67)**	**3.39 (3.13, 3.67)**	**6.37 (5.85, 6.94)**	**1.54 (1.43, 1.66)**	**2.18 (2.02, 2.35)**	**2.91 (2.70, 3.13)**	**4.46 (4.12, 4.84)**

## Data Availability

Data, including spatial exposure estimates, are available to all approved UK Biobank projects.

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
