# Peer review of "Green Walkability and Physical Activity in UK Biobank: A Cross-Sectional Analysis of Adults in Greater London"

_ijerph, 2022, doi:10.3390/ijerph19074247_

Round 1

Reviewer 1 Report

The purpose of the current study was to associations of walkability and green walkability with physical activity in an urban adult cohort.

My review will follow the basic structure identified by IJERPH as Editorial Criteria. Please note that the following comments are not intended to degrade or break the spirit of the researchers, but are provided in an attempt to improve the quality of this work.

The topic of this study is both topical and worthy of investigation. Green space seem to be associated with many beneficial health effects, including reduced all-cause and cardiovascular mortality and improved mental health, possibly through mediators such as reduced air pollution, temperature and stress and increased physical activity, social contacts and restoration. Although this field is evolving rapidly and scientific work has been carried out in recent years, there are still questions concerning the links between green spaces and the practice of physical activities and sports. The present paper discusses the concept of walkability. This concept is important in sustainable urban design and has many health, environmental, and economic benefits. The objective of this study is to explore the relationship between green space and walkability, and their combined impact on physical activity by incorporating green or vegetation cover into a walkability index. De plus, to my knowledge, the integration of high-resolution data on vegetation into a walkability index has not previously been attempted.

The authors conceptualize their work by addressing key questions and issues. not being a specialist in walkability, I will incorporate a clear and precise definition of this central concept. But the introduction allows to clearly identify the objective and allows an adapted state of the art.

The authors used cross-sectional data from the UK Biobank for which participants had completed baseline lifestyle questionnaires, participated in physical tests, and provided their residential address. Moreover, a subset participated in a 7-day accelerometer-based physical activity follow-up. How were these participants selected (criteria, indicators, variables)?

Then the authors define the different variables and present the tools and data taken into account to finally present their statistical analyses. The authors used several multinomial log-linear models and linear regression models. Therefore, I suggest clarifying the objectives with respect to these different analyses at the end of the introduction. The idea would be to detail the variables taken into account and the reasons why these variables or parameters were chosen. In this way, I think the authors could make some hypothesis.

The results bring real added value on the methodological, theoretical and applied dimensions, and the exposed limits are as many perspectives and new tracks to explore.

I am not an English native speaker myself, so I rather refuse to provide detailed corrections about language.

Finally, this manuscript is likely to make a significant contribution to the literature, and clearly has potential value for researchers, city planners and more broadly for public policies, especially in large cities. The manuscript has positive attributes and I have no reservations about conceptual, methodological and interpretive dimensions.

Author Response

Response to Reviewer 1

R1.1 The purpose of the current study was to associations of walkability and green walkability with physical activity in an urban adult cohort. My review will follow the basic structure identified by IJERPH as Editorial Criteria. Please note that the following comments are not intended to degrade or break the spirit of the researchers, but are provided in an attempt to improve the quality of this work.

The topic of this study is both topical and worthy of investigation. Green space seem to be associated with many beneficial health effects, including reduced all-cause and cardiovascular mortality and improved mental health, possibly through mediators such as reduced air pollution, temperature and stress and increased physical activity, social contacts and restoration. Although this field is evolving rapidly and scientific work has been carried out in recent years, there are still questions concerning the links between green spaces and the practice of physical activities and sports. The present paper discusses the concept of walkability. This concept is important in sustainable urban design and has many health, environmental, and economic benefits. The objective of this study is to explore the relationship between green space and walkability, and their combined impact on physical activity by incorporating green or vegetation cover into a walkability index. De plus, to my knowledge, the integration of high-resolution data on vegetation into a walkability index has not previously been attempted.

The authors conceptualize their work by addressing key questions and issues. not being a specialist in walkability, I will incorporate a clear and precise definition of this central concept. But the introduction allows to clearly identify the objective and allows an adapted state of the art.

RR1.1 We thank the reviewer for this favorable summary of our study.

R1.2 The authors used cross-sectional data from the UK Biobank for which participants had completed baseline lifestyle questionnaires, participated in physical tests, and provided their residential address. Moreover, a subset participated in a 7-day accelerometer-based physical activity follow-up. How were these participants selected (criteria, indicators, variables)?

RR1.2 We have added more information about the recruitment of UK Biobank participants which participated in the accelerometer study in Section 2.3: “Between February 2013 and December 2015, participants who had provided a valid email address were randomly selected and invited to participate in an accelerometer follow-up study. UK Biobank conducted the accelerometer-based follow-up in the volunteer sample between 2013 and 2016 to objectively measure physical activity over a 7-day period; detailed methodology is described elsewhere [26]. In brief, participants were mailed wrist-worn accelerometers and asked to wear them immediately and continuously on their dominant (writing) hand for one week.”

We also added an extra line on data cleaning and quality assurance by UK Biobank researchers to Section 2.3: “Physical activity was extracted by UK Biobank researchers from 100Hz raw triaxial acceleration data after calibration, removal of gravity and sensor noise, and identification of non-wear episodes [26].”

R1.3 Then the authors define the different variables and present the tools and data taken into account to finally present their statistical analyses. The authors used several multinomial log-linear models and linear regression models. Therefore, I suggest clarifying the objectives with respect to these different analyses at the end of the introduction. The idea would be to detail the variables taken into account and the reasons why these variables or parameters were chosen. In this way, I think the authors could make some hypothesis.

RR1.3. We have expanded our aims at the end of the introduction, as suggested by the reviewer and make it clear that we are exploring both self-reported and objectively measures physical activity. We have further added our hypothesis to the introduction: “We aimed to integrate vegetation cover into a walkability index to create a green walkability index to explore the added benefit of green space exposure on self-reported and objectively measured physical activity, using UK Biobank. Our hypothesis was that green walkability has added benefits to walkability in terms of increasing physical activity.”

R1.4 The results bring real added value on the methodological, theoretical and applied dimensions, and the exposed limits are as many perspectives and new tracks to explore.

RR1.4 We thank the reviewer for this favorable statement.

R1.5 I am not an English native speaker myself, so I rather refuse to provide detailed corrections about language.

RR1.5 We have carefully checked the manuscript for any spelling and grammar issues.

R1.6 Finally, this manuscript is likely to make a significant contribution to the literature, and clearly has potential value for researchers, city planners and more broadly for public policies, especially in large cities. The manuscript has positive attributes and I have no reservations about conceptual, methodological and interpretive dimensions.

RR1.6 We thank the reviewer for this favorable summary of our study.

Reviewer 2 Report

Major concerns:

Introduction: this part should be strengthened to make a clear picture for the readers. Please identify your topic and provide essential context.

Page 4, section 2.5.3: why ground cover is “lower than 2.5meters?” We usually think ground cover as grasses and small shrubs less than 1.5meters, which cannot block people’s view. Please find the references for supporting your ideas.

Why categorize the vegetation into tree-cover and ground-cover? What is the relationship with green walkability?

Section 2.6 and the Results: please state clearly for the sample sizes and statistical methods for every part of the results. For example: add “(n= )” for “Participants with additional accelerometer information” in line 231.

Page 7 and Table 2: please state the sample sizes. The results are not presented clear: did “walkability,” “green walkability,” “ground-cover walkability,” “tree-cover walkability” put into the model separately? What the authors want to conclude from the results? The indexes are all related to physical activity, but what the differences?

It is creative to use “green walkability,” but the advantages of this indicator, its differences from other indicators, and its unique impact are not clearly explained in the manuscript.

Minors:

Line 330: should it be “less than 125mg?”

Lines 179-180: where is “Appendix?”

Table 4, line 298: should it be ”p<=0.05?”

Author Response

Response to Reviewer 2

R2.1 Introduction: this part should be strengthened to make a clear picture for the readers. Please identify your topic and provide essential context.

RR2.1 As suggested by the reviewer, we have added more context to the introduction. We added a reference in the introduction which was important in seeding our interest in tree cover versus other types of vegetation cover: “Respondents to the London Travel Demand Survey showed higher odds of walking associated with higher density of street trees surrounding their residential address (OR = 1.06, 95% CIs = 1.03, 1.10), and higher street connectivity was also associated, but total vegetation cover surrounding addresses was not associated with higher odds of walking [15].”

We additionally added a sentence on the relevance of understanding tree cover versus other vegetation for planning:

“More specifically, the different impact of trees, which can be planted along streets without substantially impacting business and residential density (i.e. components of walkability), compared to other types of vegetation (e.g. grass) is essential information for greening cities to encourage physical activity.”

R2.2 Page 4, section 2.5.3: why ground cover is “lower than 2.5meters?” We usually think ground cover as grasses and small shrubs less than 1.5meters, which cannot block people’s view. Please find the references for supporting your ideas.

RR2.2 We acknowledge the reviewer’s concern regarding defining ground cover as lower than 2.5 meters. This was however the cut-off used in the available high-resolution vegetation data. We have expanded section 2.5.3 to make it clear that in our case ground cover includes grasses, shrubs and saplings below the height of 2.5 metres: “Vegetation data was categorised by height into tree cover (equal to or higher than 2.5 meters) and ground cover (lower than 2.5 meters) (Fig. 1). In this study, ground cover is a catch-all term that we use to describe ground vegetation, shrubs, and saplings. The granularity of the available greenspace data was our principal motivation for focusing on UK Biobank participants residing in Greater London.”

R2.3 Why categorize the vegetation into tree-cover and ground-cover? What is the relationship with green walkability?

RR2.3 As indicated in our response to R2.1 by the reviewer, our motivation on differentiating between tree cover and ground cover followed previous evidence that suggested that tree cover might have a different impact on physical activity than ground cover. We have expanded the introduction accordingly: “Respondents to the London Travel Demand Survey showed higher odds of walking associated with higher density of street trees surrounding their residential address (OR = 1.06, 95% CIs = 1.03, 1.10), and higher street connectivity was also associated, but total vegetation cover surrounding addresses was not associated with higher odds of walking [15].”

R2.4 Section 2.6 and the Results: please state clearly for the sample sizes and statistical methods for every part of the results. For example: add “(n= )” for “Participants with additional accelerometer information” in line 231.

RR2.4 Following the reviewer’s suggestion, we have added sample size throughout section 2.6 to improve clarity and allow the reader to quickly relate this information to tables.

R2.5 Page 7 and Table 2: please state the sample sizes. The results are not presented clear: did “walkability,” “green walkability,” “ground-cover walkability,” “tree-cover walkability” put into the model separately? What the authors want to conclude from the results? The indexes are all related to physical activity, but what the differences?

“Walkability, green walkability, ground-cover walkability and tree-cover walkability were assessed as the independent (exposure) variable in separate models (n = 57,726 for all models).”

R2.6 It is creative to use “green walkability,” but the advantages of this indicator, its differences from other indicators, and its unique impact are not clearly explained in the manuscript.

RR2.6 We have made the importance of green walkability clear now throughout the manuscript and have particularly strengthened the introduction and discussion section. We have added references to previous studies which found a relationship between green areas, walkability and physical activity and explain why this is important for policy makers in the introduction: “Number of parks was included in the International Physical Activity and the Environment Network (IPEN) study walkability index, which was associated with walking for transport and for leisure [12], and with objectively-measured moderate-and-vigorous physical activity (MVPA) [13].” For city planners, balancing neighbourhood density with adequate access to greenspace is a challenge, as open green spaces (e.g. parks) contain few residences, destinations, and street networks, which contribute to walkability. One exception is street trees, which can be integrated into a dense, compact urban plan that supports walking. Respondents to the London Travel Demand Survey showed higher odds of walking associated with higher density of street trees around their residential address (OR = 1.06, 95% CIs = 1.03, 1.10), street connectivity was also positively associated with walking, but total vegetation cover surrounding addresses was not associated with walking [15].” “More specifically, the different impact of trees, which can be planted along streets without substantially impacting business and residential density (i.e. components of walkability), compared to other types of vegetation (e.g. grass) is essential information for greening cities to encourage physical activity.” We have further added a Findings in context section to the discussion, section 4.3.

R2.7 Line 330: should it be “less than 125mg?”

RR2.7 More than 125mg is correct.

R2.8 Lines 179-180: where is “Appendix?”

RR2.8 We thank the reviewer for spotting this inconsistency. We have changed ‘Appendix’ to ‘Supplement material’ throughout the manuscript.

R2.9 Table 4, line 298: should it be ”p<=0.05?”

RR2.9 We thank the reviewer for spotting this minor error. We have fixed this in Table 4.  

Reviewer 3 Report

This study demonstrated that both the self-reported and objectively-measured physical activity levels were associated with walkability scores. Although this study is a cross-sectional design, the data are potentially interesting and important. However, it is necessary to show the participants’ characteristics in more detail.

The participants’ characteristics of each quintile in the walkability indices have been unclear. The authors discussed the potential co-factors such as socioeconomic factors in lines 356 to 358. However, the author should indicate the basal characteristics of each quintile including household income and commuting and non-commuting transport and compare in further tables or supplemental tables, and discuss as necessary.

It is better to indicate additional basal data such as exercise habit or domains of physical activity (leisure-time, occupational, domestic and commuting). If these data don’t exist, the authors should mention in limitation paragraph or discuss.

Author Response

Response to Reviewer 3

R3.1 This study demonstrated that both the self-reported and objectively-measured physical activity levels were associated with walkability scores. Although this study is a cross-sectional design, the data are potentially interesting and important.

RR3.1 We thank the reviewer for this favorable summary of our study.

R3.2 However, it is necessary to show the participants’ characteristics in more detail. The participants’ characteristics of each quintile in the walkability indices have been unclear. The authors discussed the potential co-factors such as socioeconomic factors in lines 356 to 358. However, the author should indicate the basal characteristics of each quintile including household income and commuting and non-commuting transport and compare in further tables or supplemental tables, and discuss as necessary.

RR3.2 Following the reviewer’s suggestion, we have added Tables TS2 – TS17 to the supplement material to show participants characteristics by quintiles of walkability, green walkability, ground cover walkability and tree cover walkability for all participants and for the sub-study with accelerometer information.

R3.3 It is better to indicate additional basal data such as exercise habit or domains of physical activity (leisure-time, occupational, domestic and commuting). If these data don’t exist, the authors should mention in limitation paragraph or discuss.

RR3.3 We thank the reviewer for this suggestion. We added to the discussion a suggestion for future research on physical activity by domain (e.g. leisure-time, occupational), which was not feasible in the scope of our study, but is crucial for developing understanding of associations of the environment and physical activity: “Future analyses should explore physical activity beyond commute versus non-commute activity and may benefit from understanding differences in leisure-time, occupational and domestic physical activity in relation to environmental exposures.”

Reviewer 4 Report

I would like to thank you for the opportunity to review this article and congratulate the authors for this work. For me, as a physical educator, this topic is very important and has a lot of value. However, I suggest some improvements so that the article can be accepted for publication. The following are my suggestions.

This article analyzed the relationship between Green Walkability and physical activity level in adults.

Title: The title is not quite concrete. As a suggestion, the title should provide information on the subject group and context being studied. 

Abstract: I recommend writing the abstract again through the general rules for writing a good abstract. This is the most important section of the paper, so it needs the most attention. 

The general structure of a good abstract is as follows: first introduce the context, then the previous lines of research, then your research questions (RQ) on which this gap has been inspired. Then the methods that allow you to answer the RQs, then the main results, and finally the conclusions. A brief note on the significance of the research is an excellent ending to a high-level summary.

Introduction
As I mentioned, I find this research extremely important in contributing to the field of Physical Activity and Health. I do not disagree with the authors' justifications and read many very good and current arguments.  However it is suggested to provide more information about the conditions that should be given to consider a space as Green Walkability.

It is suggested to the authors that based on the stated objective they highlight the research questions that help to conduct the research and the discussion based on the findings in which the study variables, the study population, and the expected result appear.

Material and method.

Participants. The characteristics of the sample should be better defined. Sex, location of the locality of residence if known (rural or urban)....
It is suggested to include a section on instruments. In addition, the psychometric properties of the IPAQ should be better defined and specified; although it is a well-known instrument, this information should be provided for those readers who are unfamiliar with the instrument.
Statistical analysis: were any tests performed to analyze the assumption of normality in the distribution of the data?

Results: 
The results are displayed correctly and are easy to read and simple for a scholar not accustomed to quantitative methodology.

Discussion: This section should be rewritten, it seems more like a presentation of results than a discussion. Findings should be more thoroughly contrasted with other studies. Please give value to this section.

Limitations: It should be noted that the data used are more than six years old. 

Author Response

Response to Reviewer 4

R4.1 I would like to thank you for the opportunity to review this article and congratulate the authors for this work. For me, as a physical educator, this topic is very important and has a lot of value.

RR4.1 We thank the reviewer for this favorable summary of our study.

R4.2 However, I suggest some improvements so that the article can be accepted for publication. The following are my suggestions. This article analyzed the relationship between Green Walkability and physical activity level in adults. Title: The title is not quite concrete. As a suggestion, the title should provide information on the subject group and context being studied. 

RR4.2 Following the suggestion by the reviewer we have revised our title and provide more information on the subjects and context as follows: “Green walkability and physical activity in UK Biobank: a cross-sectional analysis of adults in Greater London”

R4.3 Abstract: I recommend writing the abstract again through the general rules for writing a good abstract. This is the most important section of the paper, so it needs the most attention. The general structure of a good abstract is as follows: first introduce the context, then the previous lines of research, then your research questions (RQ) on which this gap has been inspired. Then the methods that allow you to answer the RQs, then the main results, and finally the conclusions. A brief note on the significance of the research is an excellent ending to a high-level summary.

RR4.3 We have revisited the abstract following the reviewer’s concerns. The journal style for abstracts does not include a breakdown into background, methods, results and conclusion section. We have however followed this structure in our abstract. We clearly identify a gap in knowledge and our research question. We then present the methods to address this question. We have added a sentence on the statistical models used: “We assessed associations using log-linear and logistic regression models, adjusted for individual- and area-level confounder.” We then present the main results and highlight the significance of our work.

R4.4 Introduction. As I mentioned, I find this research extremely important in contributing to the field of Physical Activity and Health. I do not disagree with the authors' justifications and read many very good and current arguments.  However it is suggested to provide more information about the conditions that should be given to consider a space as Green Walkability.

RR4.4 We have expanded the introduction to make a stronger argument for why green walkability should be considered and also why we need to differentiate between trees and ground cover vegetations. This is based on previous findings in the literature which we have now added to the introduction: “Number of parks was included in the International Physical Activity and the Environment Network (IPEN) study walkability index, which was associated with walking for transport and for leisure [12], and with objectively-measured moderate-and-vigorous physical activity (MVPA) [13].” For city planners, balancing neighbourhood density with adequate access to greenspace is a challenge, as open green spaces (e.g. parks) contain few residences, destinations, and street networks, which contribute to walkability. One exception is street trees, which can be integrated into a dense, compact urban plan that supports walking. Respondents to the London Travel Demand Survey showed higher odds of walking associated with higher density of street trees around their residential address (OR = 1.06, 95% CIs = 1.03, 1.10), street connectivity was also positively associated with walking, but total vegetation cover surrounding addresses was not associated with walking [15].” “More specifically, the different impact of trees, which can be planted along streets without substantially impacting business and residential density (i.e. components of walkability), compared to other types of vegetation (e.g. grass) is essential information for greening cities to encourage physical activity.”

R4.5 It is suggested to the authors that based on the stated objective they highlight the research questions that help to conduct the research and the discussion based on the findings in which the study variables, the study population, and the expected result appear.

RR4.5 We have now added our hypothesis to the introduction to make the overall aim of the research clearer including more detail on the studied variables: “We aimed to integrate vegetation cover into a walkability index to create a green walkability index to explore the added benefit of green space exposure on self-reported and objectively measured physical activity, using UK Biobank. Our hypothesis was that green walkability has added benefits to walkability in terms of increasing physical activity.”

R4.6 Participants. The characteristics of the sample should be better defined. Sex, location of the locality of residence if known (rural or urban)....

RR4.6 Detailed sample characteristics including sex are presented in Table 1. We have added additional Tables TS2-TS17 to the supplement material to describe participant characteristics by quintiles of each walkability index to fully characterize the study population for each analysis. Additionally, we are focused on London, and we have made it clear in the discussion that this is a largely urban population, with access to relatively good public and active travel infrastructure compared to smaller cities and towns and rural locations.

R4.7 It is suggested to include a section on instruments. In addition, the psychometric properties of the IPAQ should be better defined and specified; although it is a well-known instrument, this information should be provided for those readers who are unfamiliar with the instrument.

RR4.7 We have carefully considered the reviewer’s comment regarding additional information on instruments and IPAQ. We have however decided that the information already provided in the text is sufficient. Measurements were undertaken by UK Biobank and described in detail in relevant protocols. We have added references to these protocols throughout. We have however added more information on the accelerometer data: “Physical activity was extracted by UK Biobank from 100Hz raw triaxial acceleration data after calibration, removal of gravity and sensor noise, and identification of non-wear episodes [26].”

R4.8 Statistical analysis: were any tests performed to analyze the assumption of normality in the distribution of the data?

RR4.8 In advance of any statistical analysis being conducted, we took due care to explore the distribution and spread of the variables and their correlation. We decided to not include this initial data exploration as to not overburden the manuscript. We are happy to share this initial data exploration with the reviewer if of interest.

R4.9 The results are displayed correctly and are easy to read and simple for a scholar not accustomed to quantitative methodology.

RR4.9 We thank the reviewer for this favorable assessment of our result section.

R4.10 Discussion: This section should be rewritten, it seems more like a presentation of results than a discussion. Findings should be more thoroughly contrasted with other studies. Please give value to this section.

RR4.10 Following the reviewer’s suggestion we have added more depth and comparison to the discussion and added the section 4.3 Findings in context to our discussion.

R4.11 Limitations: It should be noted that the data used are more than six years old. 

RR4.11 We state in the limitations that UK Biobank does not have longitudinal exposure and outcome data which limits causal interpretation. We would prefer to have the greenness data temporally aligned to 2006-2010 baseline, but this was not available.

Round 2

Reviewer 3 Report

The manuscript is satisfactory improved.

Reviewer 4 Report

I would like to thank the authors for taking my contributions into account. The manuscript has improved sufficiently and I recommend its publication.